# Current Treatment of Anterior Communicating Artery Aneurysms: Single Center Study

**DOI:** 10.3390/brainsci10080501

**Published:** 2020-07-31

**Authors:** Ondřej Navrátil, Kamil Ďuriš, Vilém Juráň, Karel Svoboda, Jakub Hustý, Evžen Hovorka, Eduard Neuman, Andrej Mrlian, Martin Smrčka

**Affiliations:** 1Department of Neurosurgery, Faculty of Medicine, Masaryk University Brno, 62500 Brno, Czech Republic; kamoo@seznam.cz (K.Ď.); juran.vilem@fnbrno.cz (V.J.); svoboda.karel@fnbrno.cz (K.S.); hovorka.evzen2@fnbrno.cz (E.H.); neuman.eduard@fnbrno.cz (E.N.); mrlian.andrej@fnbrno.cz (A.M.); smrcka.martin@fnbrno.cz (M.S.); 2Department of Neurosurgery, University Hospital Brno, 62500 Brno, Czech Republic; 3Department of Pathophysiology, Faculty of Medicine, Masaryk University Brno, 62500 Brno, Czech Republic; 4Department of Radiology and Nuclear Medicine, Faculty of Medicine, Masaryk University Brno, 62500 Brno, Czech Republic; jakub.husty@fnbrno.cz; 5Department of Radiology and Nuclear Medicine, University Hospital Brno, 62500 Brno, Czech Republic

**Keywords:** intracranial aneurysms, subarachnoid hemorrhage, anterior communicating artery, outcome, treatment

## Abstract

Introduction: Anterior communicating artery aneurysms (ACoAAs) are the most frequent intracranial aneurysms treated at neurosurgical departments with a vascular program. Material and methods: We reviewed patients with ACoAAs in a single institution over ten years (2008–2017). The focus was on the final outcome; complications, age, and clinical condition with respect to modalities were analyzed. Results: A total of 198 patients treated during this period was included in the study: 176 patients had a ruptured ACoAA and 22 had an unruptured ACoAA. Then, 127 (71%) were treated surgically and 51 (29%) by endovascular means. Out of the whole series, a good recovery occurred in 123 patients (62%), moderate disability in 11 (5.5%), severe disability in 19 (10%), vegetative state in 11 (5.5%), and death in 34 (17%). In the 157 patients (72.5%) with a subarachnoid hemorrhage (SAH), both modalities had a favorable outcome: 27.5% had an unfavorable outcome, 12% had complications in surgery versus 17.6% during endovascular treatment. No statistical difference in outcome, complications, and age was noted between modalities. Surgical treatment was more frequently adopted for patients in a better clinical condition (*p* ≤ 0.05). Conclusion: More than two thirds of the patients (72.5%) reached a favorable outcome. There was no difference in age between the treatment modalities. Risks of complications are present and specific for both modalities.

## 1. Introduction

Anterior communicating artery aneurysms (ACoAAs) are the most frequently occurring intracranial aneurysms. The proportion of these aneurysms is about 37% [1]. Anatomically, the segment of the anterior communicating artery is variable and located deep between both frontal lobes, which makes the microsurgery of these aneurysms even more demanding [2]. The treatment of intracranial aneurysms requires teams composed of both microsurgical and endovascular members as well as dedicated intensive care units and rehabilitation centers. Despite the advances in both treatment modalities, the treatment of these aneurysms remains challenging. We reviewed our institutional algorithms of treatment, outcome, and complications of both methods.

## 2. Material and Methods

The Department of Neurosurgery, University Hospital Brno at Masaryk University, Brno, Czech Republic, provides neurosurgical services to southern Moravia and adjacent areas with a population of about 1.8 million. The hospital serves as a center of excellence in the healthcare system of the Czech Republic in this region. Our department is one of two neurosurgical centers in this region. The neurosurgical patients are referred to the hospital from regional hospitals or through the Emergency Department at the Brno University Hospital. The majority of the patients with a spontaneous subarachnoid hemorrhage (SAH) and brain aneurysms from the catchment area are treated at the Department of Neurosurgery, University Hospital Brno at Masaryk University, Brno, Czech Republic.

Since 2008, the patients’ data have been recorded in the aneurysm and subarachnoid hemorrhage database. All the patients with a non-traumatic subarachnoid hemorrhage, with a diagnosis of a cerebral aneurysm or perimesencephalic subarachnoid hemorrhage and incidental aneurysms are recorded.

### 2.1. Treatment of the Aneurysm

At the department, patients with a spontaneous subarachnoid hemorrhage (SAH) are admitted to the neurosurgical intensive care unit (ICU), where the acute stage of the treatment is carried out. After admission and initial stabilization, patients are always assessed in relation to aneurysm treatment modality. The clinical condition of a patient (World Federation of Neurosurgical Societies grade—WFNS grade, Hunt–Hess grade) on admission plays a major role. In our institution, the patients with an aneurysmal SAH are treated actively such that the aneurysm is secured within 24 h of admission, the patient receives an external ventricular drain if needed, and other necessary medical procedures and treatments are performed. If a patient is in the WFNS grade 1–5 with both pupils equal and reactive without significant brain ischemia on a computed tomography (CT) scan, we treat the patient actively. However, patients with a poor grade and bilateral mydriasis or significant brain ischemia are managed conservatively—the aneurysm is not secured and the local protocols for poor grade patients are followed. Occasionally, a patient who was initially deemed unsalvageable but subsequently shows marked clinical improvement (sometimes after the insertion of external ventricular drainage), may be reconsidered for an active approach.

The initial decision about treatment modality in a SAH is based on the clinical assessment, anatomical features of the aneurysm, the presence of an intracerebral hematoma, age, blood thinners in current medication, and associated comorbidities. The decision-making in the acute stage of a subarachnoid hemorrhage is described in the protocol in Figure 1. The aneurysm treatment is assessed by the vascular neurosurgeon and endovascular radiologist and the procedure itself is performed by experienced surgeons from both teams.

The projection of the aneurysm dome of an ACoAA is an important factor in selecting the suitable treatment modality. If the main axis of the aneurysm is projecting anteriorly or inferiorly from anterior communicating artery (ACom) (in front of or below both A2 segments), we prefer microsurgery as the first option for such patients. In these cases, the aneurysm is relatively accessible by open microsurgery; it projects toward the corridor of the surgeon’s access to the region, points away from ACom perforators and does not require the excessive retraction and resection of the ipsilateral frontal lobe. In our experience, splitting the interhemispheric fissure and at least the partial resection of the gyrus rectus in an acute SAH is always advantageous for enhancing the view of the operative field. Conversely, if the aneurysm is projecting posteriorly from these two border positions, we prefer endovascular treatment as the first option. In these cases, the aneurysm may be surrounded by hypothalamic perforators, requiring the difficult dissection and additional retraction of the frontal lobe. If the aneurysm is projecting between both A2 branches, clipping may be challenging, so the modality is chosen on an individual basis.

We involve patients with an unruptured aneurysm more in the decision-making process. However, both modalities can only be used if they are suitable for the given aneurysm. Factors for consideration during the decision-making stage of aneurysm treatment are presented in Figure 1.

### 2.2. Treatment in ICU after Aneurysm Occlusion

After the aneurysm is secured either by microsurgery or by endovascular means, the treatment in the ICU plays a key role in the success of the previous procedure. Initially, some patients require sedation; the main factors in the decision being: a poor grade, having undergone a complicated procedure, post-operative CT scans, and transcranial Doppler (TCD) velocities. The weaning of the patient is planned on an individual basis. Daily treatments/procedures include: routine postprocedural CT scanning, hydrocephalus management, transcranial Doppler measurement, blood pressure and intracranial pressure (ICP) management, ensuring sufficient fluid and nutrient intake and ion balance, physiotherapy, pressure and stress ulcer prevention, and deep venous thrombosis prophylaxis. The most important factor in avoiding the threatening condition during ICU management—delayed cerebral ischemia—is the management of systolic blood pressure (SBP). This has to be maintained individually, but at a high enough level which is usually between 150 and 70 mmHg of SBP or sometimes even higher, up to 190 mmHg SBP in some patients depending on the severity of the vasospasm and clinical status.

After the acute phase, usually between days 12 and 21 post-bleed, patients in a stable condition with secured aneurysms are discharged or transferred to the referring regional hospital. Out of the whole duration of their stay in the neurosurgical department, the patients only spend the minimum necessary time in the ICU (based on their clinical status); the rest is spent in a general ward.

### 2.3. Outcome Assessment

The patients are assessed with the Glasgow Outcome Scale after three months, either at the outpatient clinic or they or the referring department are contacted by phone.

### 2.4. Statistical Evaluation

The basic characteristics of the patients are presented in absolute (n) and relative (%) values as well as by means of descriptive statistics (mean, +/−SD). The data distribution was tested using the D’Agostino and Pearson normality test. The statistical analyses were performed by unpaired t-tests with Welch’s correction and Chi-squared tests for trend. A result of *p* < 0.05 was considered as statistically significant.

## 3. Results

During the years 2008–2017, we treated 675 patients with an intracranial cerebral aneurysm in all locations. These patients were recorded in our institutional database. Among those, 198 patients were treated with ACoAA and were included in this study. In total, 108 were men and 90 were women. The mean age was 53 years (20–89 years). Further, 176 patients had a ruptured ACoAA and 22 an unruptured ACoAA, while 127 (71%) patients were treated surgically and 51 (29%) by endovascular means. Twenty patients were not treated, because they were in a poor condition.

Among those treated surgically (127 patients), 114 had a ruptured ACoAA and 13 had unruptured aneurysms. In addition, 123 (97%) patients underwent surgical clipping and four patients (3%) had their aneurysm wrapped due to the unfavorable anatomical situation or the fusiform nature of their aneurysm, which precluded surgical clipping. Out of a further three cases from the surgically treated patients, one had their aneurysm clipped after an unsuccessful endovascular attempt at treatment, the second had their aneurysm wrapped after an unsuccessful endovascular attempt at treatment and the third had their aneurysm clipped due to aneurysm regrowth after previous coiling of the same aneurysm.

Among patients treated by endovascular means (51 patients), 43 patients had a ruptured aneurysm, eight had an unruptured aneurysm, and six required further treatment for their aneurysm due to their regrowth. In 49 patients, coiling was performed. In two cases the aneurysm was treated by stenting and coiling. One patient had endovascular treatment for aneurysm regrowth and rebleeding after previous clipping.

Over the years, the proportion of patients treated by endovascular treatment has grown but has still not reached the number of patients treated by microsurgery. The proportion of both modalities for the period is depicted in Figure 2.

The outcome of the whole series of 198 patients with an ACoAA after three months was good recovery in 123 patients (62%), moderate disability in 11 (5.5%), severe disability in 19 (10%), vegetative state in 11 (5.5%), and death in 34 (17%) (Table 1). This includes not only the SAH patients from ACoAA, but also elective cases and patients with a poor grade not undergoing any procedure for the aneurysm treatment. All the poor grade patients died.

The 157 patients with a SAH with ACoAA, treated actively irrespective of the treatment modality (includes both microsurgery and endovascular treatment), had a high likelihood of reaching a favorable outcome (72.5%). Only 27.5% had an unfavorable outcome (Table 2). In this table, poor grade patients not amenable to treatment and elective procedures were excluded.

Out of the patients with a surgically treated aneurysm and SAH (114 patients), 77 (68%) experienced a good recovery, 9 (8%) moderate disability, 12 (10%) severe disability, 9 (8%) vegetative state, and 7 death (6%). Further, 75.5% of the patients in the surgical group achieved a favorable outcome and 24.5% had an unfavorable outcome (Table 3).

In the endovascular group with SAH from ACoAA, 27 (63%) patients experienced good recovery, 1 (2%) moderate disability, 6 (14%) severe disability, 2 (5%) vegetative state, and 7 (16%) died. A favorable outcome was achieved in 65% and an unfavorable outcome in 35% of the patients (Table 4). The difference in outcomes between the surgery and endovascular groups is not significant (*p* = 0.165, Fisher’s exact test, Figure 3).

Complications were present in both modality groups. Among the patients treated microsurgically, 15 patients (12%) had complications related to the treatment for the aneurysm, eight (6.27%) patients experienced intraoperative aneurysm ruptures, and three (2.4%) experienced rebleeding (one patient with a fatal outcome, one with an unfavorable outcome, and one patient was reclipped with a favorable outcome). In three patients (2.4%), a recurrent artery of Heubner ischemia was found postoperatively. One patient (0.8%) had an intraoperative perforation of the internal carotid artery (ACI) during dissection without sequelae. In the endovascular group, nine (17.6%) patients had complications related to the treatment for the aneurysm, six experienced intraprocedural thromboembolic events, two (3.9%) patients had intraprocedural aneurysm ruptures (one with a fatal outcome and one without consequences with a favorable outcome). One (2%) patient had coil migration without consequences. Statistically, there was no significant difference in complications between the treatment methods of surgery vs. endovascular (*p* = 0.322, Fisher’s exact test, Figure 4).

All patients treated electively experienced a good recovery. Eight patients had endovascular treatment and fourteen patients had surgical treatment (Table 5).

The mean age of all the patients undergoing the procedure for an ACom aneurysm was 52.88 (+/−12.77) in surgically treated patients and 51.32 (+/−12.15) in endovascular patients; the difference was not statistically significant (*p* = 0.45, unpaired t-test with Welch’s correction).

The mean Hunt–Hess grade (HH) was 2.22 (+/−1.06) in the group treated by microsurgery and 2.77 (+/−1.31) in the endovascular group. The difference between these groups was statistically significant (*p* = 0.02, unpaired t-test with Welch’s correction). Table 6 and Figure 5 show the proportion of surgical and endovascular treatments in the HH 1–2, HH 3, and HH 4–5 groups. The trend in treatment was statistically significant among the groups (*p* = 0.005, Chi-squared test for trend). Surgical treatment was more than three times more frequent in the HH 1–2 group, while in the HH 4–5 group, endovascular treatment was almost as frequent as surgical treatment.

## 4. Discussion

Patients with a SAH and ruptured ACoAA have a high likelihood of achieving a favorable outcome; this occurred in 72.5% of patients, irrespective of treatment modality and rupture status. Heit et al. found that at three months, 78% of ACoAA patients presenting with a SAH were equal or lower than two on the Modified Rankin Scale (mRS) in both modalities [3]. In our study, a favorable outcome was achieved in 65% of the endovascular group and in 75.5% of the surgical group. However, this difference in outcomes is not statistically significant (Figure 3). Steklacova et al. reported comparable outcomes to our cohort: a favorable outcome in 74% of the clipped patients and 70% of the coiled patients in a large single center study covering the recent period of seventeen years [4]. Pietrantonio reported better clinical outcomes in their cohort for endovascular treatment [5]. In a subgroup of the Barrow Randomized Aneurysm Trial (BRAT) patients with ACom aneurysms, there was no statistical difference in outcome between surgery and endovascular treatment [6]. In a meta-analysis by Li, which covered all intracranial aneurysms, better results were presented for endovascular methods as opposed to surgery [7]. However, the GOS and mRS scale are too broad and do not capture the subtle and subclinical deficits which can be revealed by detailed neuropsychological examination, which was not the subject of this study. Our study was not focused on neuropsychological sequelae. As reported in the International Subarachnoid Aneurysm Trial (ISAT) cognitive outcomes, approximately one-third of patients who were not disabled in mRS had cognitive impairment which was more frequent in the surgical group. This can be explained by the retraction of the frontal lobe and the gyrus rectus dissection in the anterior communicating region [8]. Among the major factors influencing the final outcome, we consider the deleterious effect of the SAH and additionally the secondary deleterious effect of delayed cerebral ischemia associated with cerebral vasospasm, considered a predominant cause of an unfavorable outcome. Angiographic vasospasm can be seen in as many as 70% of patients after SAH and 20–40% have clinical symptoms. However, the final outcome is influenced by many factors [9].

As expected, there was a difference in the distribution of the Hunt–Hess grades between both modalities. The mean Hunt–Hess grade for microsurgery was significantly lower than that for endovascular treatment (*p* = 0.02), confirming our preference for the endovascular treatment of patients in a worse clinical condition. In a recent study from Thailand, there was no statistical difference in Hunt–Hess grades and WFNS grades between surgical clipping and endovascular treatment of ruptured posterior communicating artery aneurysms [10]. A similar policy to ours—the preference to microsurgery in lower Hunt–Hess grades (1–3)—can be found in other centers [4].

Our institution has adopted the protocol of the active treatment of ruptured intracranial aneurysms. Therefore, all ruptured aneurysms amenable to treatment are addressed within 24 h of admission. This approach allows us to reduce the occurrence of rebleeding of an aneurysm and potentiate the prevention and treatment of secondary ischemia associated with cerebral vasospasm. Close 24/7 cooperation between a vascular neurosurgeon and an endovascular team is crucial for selecting and performing the appropriate treatment of the aneurysm and for combining both modalities whenever needed. In patients with two aneurysms or an aneurysm with several lobes in the ACoAA region, it is sometimes even beneficial to combine both modalities: occlude one aneurysm in this region or its lobe by endovascular means and the other by microsurgery or vice versa. The gap between the proportion of patients treated by microsurgery, which was dominant in the early years, and endovascular treatment has been steadily closing over the years. This trend is apparent in Figure 2. In the last year of this cohort, the number of patients treated by endovascular treatment was still lower than the number of patients treated by microsurgery, but their numbers were almost equal. This trend may have been caused by both institutional and global trends and influences.

In the treatment of SAH patients we follow the protocol described in Figure 1. This protocol has been developing over the years and is described here in a simplified version. Generally, we adhere to it in the vast majority of cases. Occasionally, based on an individual situation, we may decide which factor plays a more important role (e.g., in patients on blood thinners with anteriorly pointing aneurysm we have to choose endovascular treatment whereas without this medication we would opt for surgery). The protocol has proved effective over the years and will definitely be developed further in the future based on the availability and development of both treatment methods.

It is well known that the shape and size of the aneurysm, including the neck to dome ratio, play an important role in the ability of endovascular methods to occlude an aneurysm permanently [11]. The same applies to the subgroup of anterior communicating artery aneurysms. Aneurysms with broad necks, very small aneurysms, and branching vessel(s) at the aneurysm are considered unfavorable for simple embolization, and much like dual antiplatelet therapy, require additional methods to support the occluding material within the aneurysm (stenting, remodeling, etc.). The shape of the dome, including irregularities, plays a role in the behavior of the aneurysm. Irregular domes have unstable walls and are prone to rupture [12]. In the literature, surgery has been shown to offer higher stability along with a lower need for retreatment in comparison to endovascular methods [13]. Aneurysms treated by endovascular treatment have more frequent rebleeding and require more frequent retreatments than surgery [7].

Age plays an important role in undergoing the procedure and recovery from it. Elderly patients have more fragile brain tissue which is less tolerant to microsurgery even when the brain is handled gently to minimize the traumatic effect of surgery. Based on this, we tend to recommend surgical treatment for younger patients if the technical and other aspects of the treatments are favorable. In older patients (≥60 years) with multiple comorbidities and/or on blood thinners, we generally recommend endovascular treatment. Interestingly, we expected the endovascular group to be older. In reality, the age distribution of the patients in both modalities of this study was similar and the difference was statistically insignificant (*p* = 0.45). Although we tend to propose endovascular treatment for older patients, other factors, which play a role in the decision about treatment modality (Table 1), outweigh the age factor. In the literature, there was an apparent association between increasing age, after the age of 50, and a worse outcome [14]. A recent paper by Chernysev reported that endovascular treatment in the elderly provides a better clinical outcome but microsurgical clipping yields higher complete aneurysm occlusion [12].

Each treatment modality—surgery and endovascular treatment—has its specific complications related to aneurysm occlusion. Complications occurred in 12% of the surgical group and 17.6% of the endovascular group, however, this difference lacks statistical significance. This corresponds to the outcome, which is slightly worse in the endovascular group.

Recently, it has been documented that the mortality rate in SAH treated patients has decreased over time [13]. Despite the advancements of the modern neurosurgical era such as microscopes and other adjuncts (e.g., indocyanine-green videoangiography), further advances in the management of an ACoAA can still be made in endovascular methods and microsurgical techniques as well as in intensive care treatment [15]. High-level targeted neurointensive care treatment is essential for overcoming or minimizing possible acute stage complications which may lead to neurological deterioration and an unfavorable outcome.

The limitations of this study are given by its design: it is a single center and retrospective study.

## 5. Conclusions

ACoAAs are the most common intracranial aneurysms. However, their treatment remains challenging, especially when they rupture. In this single center study, more than two-thirds of the patients (72.5%) treated actively for a SAH from an ACoAA reached a favorable outcome. There was no difference in the age of patient subjects to each treatment modality. Microsurgery was performed on patients with lower Hunt–Hess grades. Risks of complications are present and specific for both modalities despite the efforts to minimize them.

## Figures and Tables

**Figure 1 brainsci-10-00501-f001:**
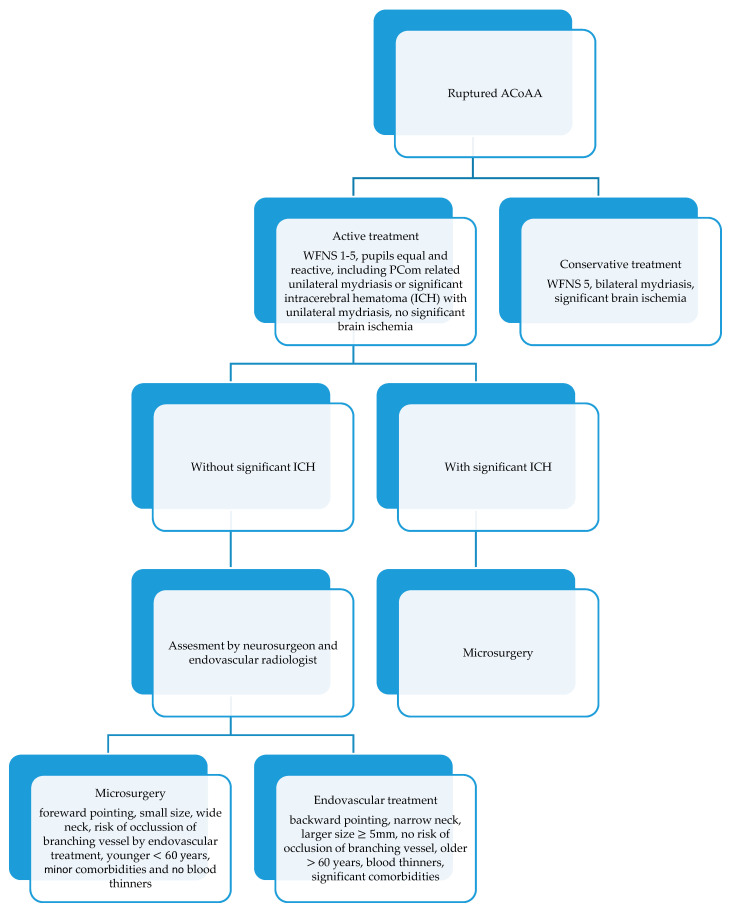
Management of ruptured anterior communicating artery aneurysms (ACoAA) in the center-simple algorithm.

**Figure 2 brainsci-10-00501-f002:**
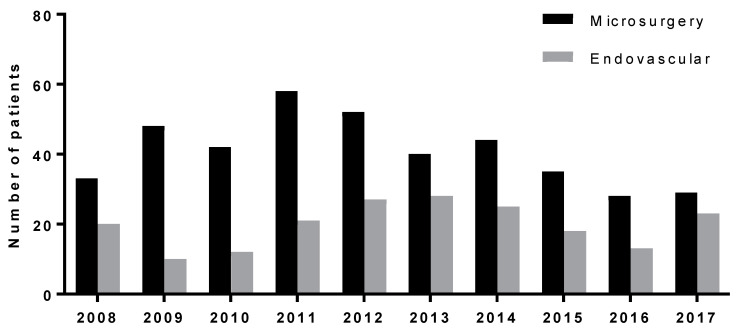
Proportion of both modalities for the period 2008–2017.

**Figure 3 brainsci-10-00501-f003:**
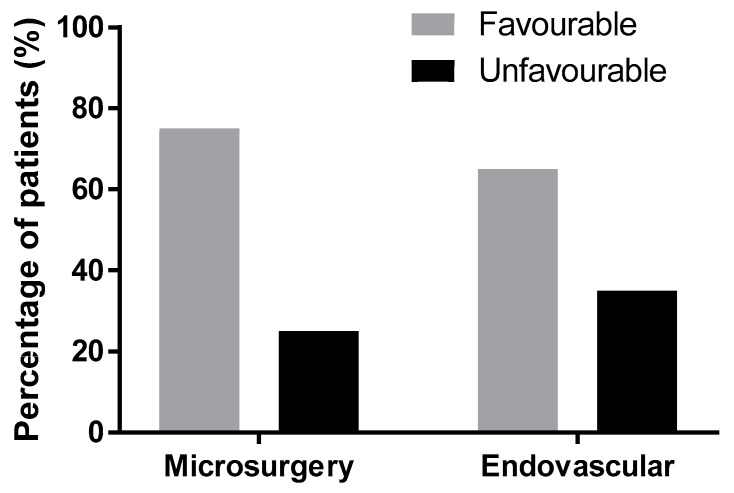
The difference between outcomes of surgery and endovascular treatment lacks statistical significance (*p* = 0.165); Glasgow Outcome Score after three months.

**Figure 4 brainsci-10-00501-f004:**
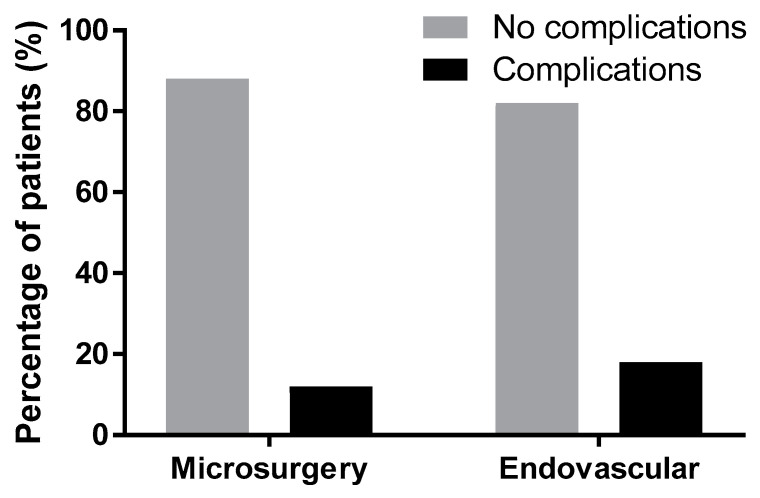
Statistical evaluation of complications in both modalities was not significant (*p* = 0.322); Glasgow Outcome Score after three months.

**Figure 5 brainsci-10-00501-f005:**
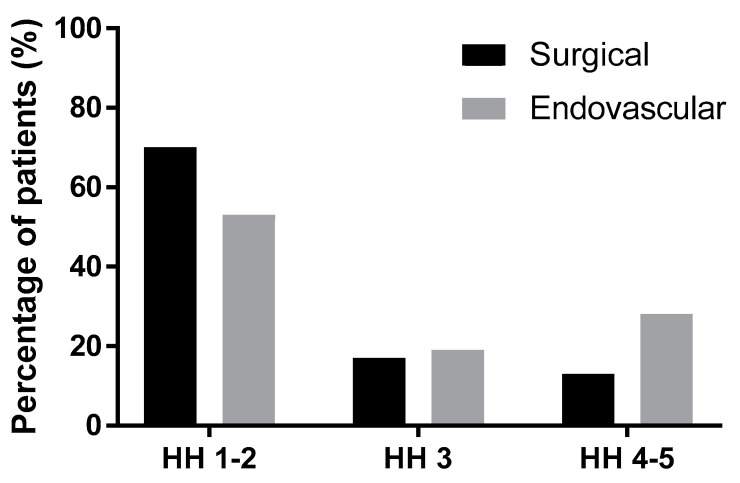
The modality of treatment according to Hunt–Hess grade (HH).

**Table 1 brainsci-10-00501-t001:** Results of treatment in the series for all patients; Glasgow Outcome Score (GOS) after three months.

Glasgow Outcome Score	No. of Patients *n* = 198
Good recovery	123 (62%)
Moderate disability	11 (5.5%)
Severe disability	19 (10%)
Vegetative state	11 (5.5%)
Death	34 (17%)

**Table 2 brainsci-10-00501-t002:** Outcome of patients with subarachnoid hemorrhage with ACoAA; GOS after three months.

Glasgow Outcome Score	No. of Patients *n* = 157	Favorable/Unfavorable Outcome
Good recovery	104 (66%)	114 (72.5%)
Moderate disability	10 (6.5%)	
Severe disability	18 (11.5%)	43 (27.5%)
Vegetative state	11 (7%)	
Death	14 (9%)	

**Table 3 brainsci-10-00501-t003:** Surgical treatment outcome of patients three months after subarachnoid hemorrhage (SAH) from ACoAA; GOS after three months.

Glasgow Outcome Score	No. of Patients *n* = 114	Favorable/Unfavorable Outcome
Good recovery	77 (67.5%)	86 (75.5%)
Moderate disability	9 (8%)	
Severe disability	12 (10.5%)	28 (24.5%)
Vegetative state	9 (8%)	
Death	7 (6%)	

**Table 4 brainsci-10-00501-t004:** Endovascular treatment outcome of patients three months after SAH from ACoAA; GOS after three months.

Glasgow Outcome Score	No. of Patients *n* = 43	Favorable/Unfavorable Outcome
Good recovery	27 (63%)	28 (65%)
Moderate disability	1 (2%)	
Severe disability	6 (14%)	15 (35%)
Vegetative state	2 (5%)	
Death	7 (16%)	

**Table 5 brainsci-10-00501-t005:** Elective treatments in ACoAA patients; Glasgow Outcome Score after three months.

Modality	No. of Patients, *n* = 22	Outcome
Endovascular (primary or redo coiling)	8 patients (36% of the elective cases)	good recovery *n* = 8 (100%)
Surgery	14 patients (64% of the elective cases)	good recovery *n* = 14 (100%)

**Table 6 brainsci-10-00501-t006:** The modality of treatment according to Hunt–Hess grade (HH).

Hunt-Hess Grade	Surgical Treatment	Endovascular Treatment
HH 1–2	78 patients (70%)	23 patients (53%)
HH 3	19 patients (17%)	8 patients (19%)
HH 4–5	14 patients (13%)	12 patients (28%)

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
