# Peer review of "Current Treatment of Anterior Communicating Artery Aneurysms: Single Center Study"

_brainsci, 2020, doi:10.3390/brainsci10080501_

Round 1

Reviewer 1 Report

- This is a well written single center retrospective series of surgical and endovascular treatment outcomes of anterior communicating artery aneurysm. 

- Cases are from a high volume center, methods are well described and results are outlined clearly. 

  • Selection criteria based on anatomic orientation is interesting.
  • Manuscript can be enriched by adding cognitive outcomes if available. If this data is not available, I would consider discussing the cognitive outcomes of ISAT in the discussion section. 
  •  To my knowledge, there is no randomized data to support not utilizing surgery in the elderly population (above age 70). In fact, results of ISAT signaled benefit to surgery over endovascular treatment in elderly patients. I would revisit this topic in the discussion section and try to discuss the current evidence. 

Author Response

Dear reviewer,

thank you very much for your valuable comments. We think that thanks to that, the quality of the manuscript improved further. The changes in the manuscript were marked in green. We have done the revisions accordingly as follows:

  • We have not evaluated the cognitive outcome, however the discussion was enriched further in regard to the cognitive outcomes not only in ISAT but other studies such as BRAT cognitive outcomes in ACoA etc. pPease see the lines 250-271
  • the discussion about the treatment of cerebral aneurysms in elderly was added, please see the lines 327-330

Additional proofreading by the native speaker was performed.

Many thanks,

Ondrej Navratil.

Reviewer 2 Report

This manuscript presents ten years of data on ACoAA in a single center. The idea behind the manuscript seems to be to determine whether or not changes need to be made to the treatment protocol. The data is somewhat different from literature on outcomes for the two groups. This may be for several reasons, and depending on the reason, may warrant a potential change in the center’s protocol for endovascular treatment. The center has good data and outcomes in microsurgery treatment and should highlight this part of the protocol is excellent (i.e. better outcomes for these patients than literature, although that depends if the same patient subcohort is used such as HH score). Additional data should be provided to determine whether or not the claims made are supported. Depending on the new data, the paper should detail what is good and what needs changing about their protocol. This manuscript would benefit if it is able to identify areas of weakness in the protocol.  This may prompt other centers to do the same.

Comments

Minor

  • In the sentence: “157 patients (72.5%) with a SAH, both modalities, had a 20 favourable outcome – 27.5% had an unfavourable outcome.” What does both modalities refer to? Surgery and treatment? If so, you should say this.
  • All supplemental material should be moved into the manuscript. The tables and figures in the supplement are important and should be in the manuscript.
  • Figure 3 has black color as favorable and grey and unfavorable, but then in figure 4 the black color is not good (it is complications). Please be consistent. If black is “good” in fig 3, then make it “good” in fig 4.
  • The data in figures 3-5 should be plotted as percentages not total number. The statistics should then be performed on the percentages (not the numbers) since the numbers are not equal between microsurgery and endovascular surgery.
  • Is “diagram 1” suppose to be “figure 1”? if so, please make this change.
  • Line 104 – “sometimes even higher” please give the entire range that is used in the dataset presented or give the quartiles and min/max.
  • Please confirm that all stable patients are discharged 14-21 days later. In our ICU, many stable patients can be discharged well before 14 days.
  • Are outcomes assessed during ICU stay? The way it is written, it does not look like it.
  • Tables 2, 3, and 6 do not say when the outcomes were measured. Based on the methods it seems like 3 months, but this needs to be written into the table legend.
  • Lines 202-204 – does this sentence refer to the dataset in this manuscript? If so, it would be beneficial to see the data on neurological assessment during ICU stay and DCI incidence for the groups. Otherwise it is difficult to conclude that the difference in outcome in this study versus other studies was due to SAH injury and DCI. There could be many other factors.
  • Lines 206-208 – how does this compare to other centers?
  • Lines 251-252 – please report on the comorbidities in the two groups to confirm this speculation.
  • the paper should detail what is good and what needs changing about their protocol.
  • Minor grammar errors need to be fixed.
  • Please describe the treatment terms: protocol for active treatment and protocol for conservative treatment.

Author Response

Dear reviewer,

thank you very much for your valuable comments. We think that thanks to that, the quality of the manuscript improved further. The changes in the manuscript during the revision were marked by green colour. We have done the revisions as follows:

  • In the sentence "In 157 patients" both modalities were added and explained, please see the lines 173-176
  • All supplemental material was moved into the manuscript and was revised - percentage in figures 3-5, GOS after three months, colour in graphs to be consistent as required.
  • diagram 1 was renamed as figure 1 - as required
  • the sentence including the statement sometimes even higher was changed and the whole sentence was improved including the entire range, please see the lines 111-114
  • we have slightly changed and explained the stay at ICU - please see the lines 115-118
  • the outcome for the purpose of this study is not assessed at ICU
  • in all the tables, the time of assessment - GOS after three was added
  • the discussion in first paragraph including the final outcome was modified according to remarks of the reviewer. The neuropsychological assessment was performed and the discussion including the results of other studies was added and disscused - ISAT, BRAT subgroup of ACoA aneurysms etc.
  • the discussion about Hunt-Hess grading and the indications to surgery vs. endovascular treatment was added as required - please see the lines 278-284.
  • speculative statement was removed and corresponding part corrected as follows: "Complications occurred in 12% of the surgical group and 17.6% of the endovascular group, however, this difference lacks statistical significance. This corresponds to the outcome, which is slightly worse in the endovascular group." - please see the lines 336-339

  • we have devoted the separate paragraph to the treatment protocol itself - figure 1 - see the lines 302-308. The pros and cons of the protocol were discussed and the view for the protocol change in the future as well 
  • The overall language and text proofreading was performed by a native speaker.
  • The terms active and conservative treatment were explained in the text, please see the paragraph starting in the line 55

Many thanks,

Ondrej Navratil